# Cyclodextrin-Mediated Cholesterol Depletion Induces Adiponectin Secretion in 3T3-L1 Adipocytes

**DOI:** 10.3390/ijms241914718

**Published:** 2023-09-28

**Authors:** Yu-Ting Chiang, Ying-Yu Wu, Yu-Chun Lin, Yu-Yao Huang, Juu-Chin Lu

**Affiliations:** 1Department of Physiology and Pharmacology, College of Medicine, Chang Gung University, Taoyuan 33302, Taiwan; 2Graduate Institute of Biomedical Sciences, College of Medicine, Chang Gung University, Taoyuan 33302, Taiwan; 3Division of Endocrinology and Metabolism, Department of Internal Medicine, Chang Gung Memorial Hospital at Linkou, Taoyuan 33305, Taiwan

**Keywords:** adipocytes, adiponectin, cholesterol, endosome/lysosome

## Abstract

Adipocytes store a significant amount of cholesterol and triglycerides. However, whether cholesterol modulates adipocyte function remains largely unknown. We modulated the cholesterol level in adipocytes to examine its effect on the secretion of adiponectin, an important hormone specifically secreted by adipocytes. Treating differentiated 3T3-L1 adipocytes with 4 mM methyl-β-cyclodextrin (MβCD), a molecule with a high affinity for cholesterol, rapidly depleted cholesterol in adipocytes. Interestingly, MβCD treatment increased adiponectin in the medium without affecting its intracellular level, suggesting a modulation of secretion. By contrast, cholesterol addition did not affect adiponectin secretion, suggesting that cholesterol-depletion-induced intracellular cholesterol trafficking, but not reduced cholesterol level, accounted for MβCD-induced adiponectin secretion. MβCD-induced adiponectin secretion was reduced after 10 μg/mL U18666A treatment that suppressed cholesterol transport out of late endosomes/lysosomes. Depleting Niemann–Pick type C1 (NPC1) or NPC2 proteins, which mediate endosomal/lysosomal cholesterol export, consistently reduced MβCD-induced adiponectin secretion. Furthermore, treatment with 1 μM bafilomycin A1, which neutralized acidic endosomes/lysosomes, also attenuated MβCD-induced adiponectin secretion. Finally, MβCD treatment redistributed cellular adiponectin to lower-density fractions in sucrose gradient fractionation. Our results show that MβCD-mediated cholesterol depletion elevates the secretion of adiponectin, highlighting the involvement of endosomes and lysosomes in adiponectin secretion in adipocytes.

## 1. Introduction

Adipocytes are known to store a significant amount of non-esterified cholesterol in their lipid droplets [1,2]. It has been hypothesized that adipose tissue serves as a cholesterol sink, and cholesterol levels can comprise from 25% of the total body cholesterol content in lean individuals up to 50% in obese individuals [3]. The abundant amount and the unique storage form of cholesterol in adipocytes suggest its role in modulating adipocyte function. Moreover, studies have shown that cholesterol levels are elevated in hypertrophied adipocytes in obesity [4], and cellular cholesterol distribution is altered in hypertrophied adipocytes [1,2,5]. Therefore, cholesterol imbalance may have an impact on adipocyte function during obesity or other diseases.

Interestingly, even though adipocytes are equipped with enzymes for cholesterol biosynthesis, the majority of cholesterol stored in adipocytes comes from circulation, particularly low-density lipoproteins (LDLs) [2,6,7]. Through LDL receptor (LDLR)-mediated endocytosis, LDL-cholesterol is brought into the cell and transported to the endo-lysosomal system. In the acidic endosome, the conformational change in the LDLR dissociates LDL-cholesterol. Finally, in lysosomes, the acid lipase hydrolyses LDL-cholesterol and exports it to other cellular locations, such as the plasma membrane and endoplasmic reticulum [8]. Defective cholesterol trafficking causes the accumulation of cholesterol in endosomes and lysosomes, leading to lipotoxicity and cell death [9,10], highlighting the importance of cellular cholesterol modulation in cell function. However, whether cholesterol modulations affect adipocyte function remains largely unknown.

Adiponectin is a hormone specifically secreted by adipocytes. It increases insulin sensitivity in peripheral tissues, such as the liver and muscle. Adiponectin is known to increase fatty acid oxidation in skeletal muscle increasing glucose uptake, thereby reducing plasma glucose levels and reducing ectopic lipid accumulation in the liver and muscle [11]. Moreover, adiponectin reduces the uptake of oxidized low-density lipoprotein and inhibits foam cell formation, thereby suppressing the development of atherosclerosis [12]. Clinically, low adiponectin levels are associated with metabolic diseases [13,14,15]. However, the secretory mechanism of adiponectin in adipocytes remains poorly characterized.

We have previously found that cholesterol depletion resulted in the disruption of lipid raft function in the plasma membrane, leading to elevated proinflammatory signaling and the production of monocyte chemoattractant protein 1 (MCP1) [16]. However, it remains unknown if cholesterol modulation in adipocytes affects the expression or secretion of other adipokines. In the current studies, we found that cholesterol depletion through MβCD treatment acutely increased adiponectin secretion from adipocytes, and endosome/lysosome function played an important role in mediating MβCD-induced adiponectin secretion.

## 2. Results

### 2.1. Depletion of Cholesterol in Adipocytes Increases Adiponectin Secretion

Adipocytes store the majority of the body’s cholesterol in non-esterified form. However, whether cellular cholesterol modulates adipocyte function remains largely unknown. Methyl-β-cyclodextrin (MβCD), a cyclic oligosaccharide with a high affinity for cholesterol, was used to rapidly remove cholesterol from both the plasma membrane and intracellular pool in adipocytes [16,17]. Differentiated 3T3-L1 adipocytes were untreated or treated with 4 mM MβCD, and total cholesterol levels were measured. The concentrations and durations of MβCD treatment did not affect cell viability or morphology (Appendix A and [16]). As shown in Figure 1A, MβCD treatment acutely reduced the cholesterol levels to 50% within an hour. Interestingly, acutely depleting cholesterol was also correlated with increased adiponectin secretion in adipocytes (Figure 1B).

Given that the MβCD-induced increase in adiponectin secretion was acute, we reasoned that MβCD treatment might modulate the secretion of adiponectin. Consistently, MβCD treatment did not affect the cellular protein level of adiponectin (Figure 1D) or its mRNA (Figure 1C).

To determine if the effects of MβCD treatment on adiponectin secretion were related to cholesterol levels, we combined the treatment of MβCD with water-soluble cholesterol (WsCL) to modulate cholesterol levels in adipocytes. WsCL, a cyclodextrin–cholesterol inclusion complex, acts as a cholesterol donor to the cells [18]. MβCD treatment reduced adipocyte cholesterol levels. In contrast, WsCL treatment increased cholesterol levels in adipocytes (Figure 2A). Moreover, combined treatment with WsCL reversed cholesterol level reductions caused by MβCD treatment (Figure 2A).

The effects of cholesterol modulation on adiponectin secretion in adipocytes were examined (Figure 2B). While MβCD treatment induced adiponectin secretion, the combined treatment of WsCL and MβCD, which reversed the reduction in cholesterol via MβCD treatment, attenuated MβCD-induced adiponectin secretion (Figure 2B). Interestingly, the addition of WsCL, which increased cholesterol levels in adipocytes (Figure 2A), did not affect adiponectin secretion (Figure 2B). Neither MβCD nor WsCL treatment affected *Adipoq* mRNA in adipocytes (Appendix A).

### 2.2. U18666A Treatment Reduced Adiponectin Secretion in Adipocytes

The depletion of cholesterol via MβCD treatment increased the secretion of adiponectin. By contrast, the addition of cholesterol via WSCL treatment did not reduce adiponectin secretion in adipocytes, suggesting that the MβCD-mediated induction of adiponectin secretion might not be due to the change in the cholesterol level, but more likely be due to altered intracellular trafficking of cholesterol caused by MβCD treatment. It has been reported that cyclodextrin treatment induced calcium-dependent lysosomal exocytosis [19], which prompted us to test if altered cholesterol modulation in lysosomes play a role in MβCD-induced adiponectin secretion.

Niemann–Pick disease type C (NPC) proteins are lysosomal proteins modulating cholesterol trafficking out of lysosomes [10]. Soluble protein NPC2 transfers cholesterol to NPC1, a transmembrane protein in late endosomes and lysosomes, facilitating cholesterol export from late endosomes and lysosomes. Defective or deficient NPC function results in the accumulation of cholesterol and sphingolipids within the late endosomal/lysosomal compartment [9,20]. To test if cholesterol trafficking from endosomes and lysosomes was involved in adiponectin secretion, we applied U18666A, a selective inhibitor of NPC1 [21,22], in adipocytes. Treatment with 10 μg/mL U18666A [23] for 2 h resulted in a 40% reduction in adiponectin secretion (Figure 3A) without affecting cellular adiponectin protein (Figure 3B) or mRNA (Appendix A), suggesting an effect on secretion. By contrast, chronic treatment (24 h) with U18666A for 24 h reduced both cellular adiponectin and its secretion (Figure 3C,D). We further examined if U18666A treatment affected MβCD-induced adiponectin secretion. As shown in Figure 3E, 2 h U18666A treatment reduced basal and MβCD-induced adiponectin secretion (Figure 3E).

### 2.3. Depletion of NPC1 or NPC2 Reduced MβCD-Induced Adiponectin Secretion

To further determine if NPC1 or NPC2 was involved in MβCD-induced adiponectin secretion, we depleted the endogenous expression of NPC1 or NPC2 in differentiated 3T3-L1 adipocytes (Figure 4A,C). The depletion of NPC1 slightly reduced basal adiponectin secretion, whereas the depletion of NPC2 in adipocytes did not affect basal adiponectin secretion (Figure 4B,D). Interestingly, the depletion of NPC1 or NPC2 in adipocytes reduced MβCD-induced adiponectin secretion (Figure 4B,D) without affecting adiponectin protein in the lysates (Appendix A). Since NPC proteins are involved in cholesterol efflux from endosomes and lysosomes, the reduction in MβCD-induced adiponectin secretion after NPC1 or NPC2 depletion suggests a role of cholesterol handling in endosomes and lysosomes in MβCD-induced adiponectin secretion.

### 2.4. Inactivation of Endosomal/Lysosomal Function Attenuated MβCD-Induced Adiponectin Secretion

Endosomes and lysosomes are known to maintain low intraluminal pH for their function [24]. To further confirm the involvement of endosomes and lysosomes in MβCD-induced adiponectin secretion, we applied the treatment of bafilomycin A1 (BafA1), which specifically inhibits vacuolar H^+^-ATPase, thereby neutralizing acidic organelles such as endosomes and lysosomes [22]. The inhibition of lysosomal function via BafA1 treatment also results in cholesterol accumulation in the lysosomes [22]. To determine if the inhibition of lysosomal function affected MβCD-induced adiponectin secretion in adipocytes, we treated differentiated 3T3-L1 adipocytes with BafA1 and measured adiponectin secretion in the presence or absence of MβCD treatment. As shown in Figure 5A, while MβCD treatment increased adiponectin secretion, BafA1 treatment reduced basal adiponectin secretion and attenuated MβCD-induced adiponectin secretion. Consistent with the suppression of adiponectin secretion, BafA1 treatment increased cellular adiponectin protein, suggesting accumulation of adiponectin in the adipocytes after BafA1 treatment (Figure 5B). These results indicate that inactivating endosomal and lysosomal function, either by increasing their pH or by interfering with cholesterol trafficking, attenuated MβCD-induced adiponectin secretion, highlighting a role of endosomes and lysosomes in intracellular trafficking and the secretion of adiponectin in adipocytes.

### 2.5. MβCD Treatment Alters Cellular Adiponectin Distribution in Lipid Raft Fractionation

Elevated adiponectin secretion after MβCD treatment might be due to the redistribution of cellular adiponectin to the intracellular location ready for secretion. Therefore, we performed lipid raft fractionation to test if MβCD treatment might cause altered adiponectin distribution. MβCD treatment is known to disrupt the integrity of the lipid rafts, the detergent-resistant microdomains (DRMs) of the plasma membrane enriched with cholesterol and glycosphingolipids [25], as shown by the altered distribution of lipid raft marker protein caveolin-1 (Figure 6, [16]). Interestingly, MβCD treatment also caused the redistribution of adiponectin to lower-density fractions (Figure 6).

## 3. Discussion

Adipocytes store a substantial amount of cholesterol and energy-storing triglycerides in their lipid droplets. Nevertheless, scant attention has been given to investigating the association between cellular cholesterol levels in adipocytes and their functional roles. Through the use of MβCD treatment to rapidly deplete cholesterol from adipocytes, our findings reveal that the acute application of MβCD treatment leads to the stimulation of adiponectin secretion by adipocytes. We further showed that the modulation of cholesterol egress from late endosomes and lysosomes by NPC proteins and functional endosomes/lysosomes might play a role in MβCD-induced adiponectin secretion (Figure 7). The findings from this study imply a link between cellular cholesterol modulation and the adiponectin secretory mechanism in adipocytes.

Direct observation of cholesterol imbalance in adipokine biosynthesis and secretion is limited in the literature. During adipocyte differentiation, cholesterol depletion induces mRNA of proinflammatory cytokines such as tumor necrosis factor α and interleukin 6, but not leptin in 3T3-L1 adipocytes [5]. Nevertheless, cholesterol depletion during adipogenesis may independently affect differentiation, leading to these changes. Moreover, MβCD treatment may deplete cell membrane cholesterol, thereby disrupting the function of lipid rafts, the microdomains enriched in cholesterol in the plasma membrane [16]. The disruption of lipid rafts or caveolae, which harbor a fraction of insulin receptors, may disrupt the assembly of receptors with downstream signaling molecules, thereby reducing insulin signaling in adipocytes [26,27]. Meanwhile, the disruption of lipid rafts also induces proinflammatory signaling and chemokine production [16]. Reduced insulin signaling and elevated proinflammatory signaling both affect leptin expression in adipocytes. However, whether the modulation of cellular cholesterol regulates adipokine secretion in adipocytes remains poorly characterized.

Our results showed that MβCD treatment depleted cellular cholesterol and increased adiponectin secretion in adipocytes. By contrast, the combined treatment of WSCL and MβCD restored MβCD-reduced cholesterol and attenuated MβCD-induced adiponectin secretion. Interestingly, adding WSCL alone to increase adipocyte cholesterol did not affect adiponectin secretion, suggesting that MβCD-induced adiponectin secretion might not be due to cholesterol reduction. Cyclodextrin treatment has been used to remove endo/lysosomal cholesterol accumulation in NPC mice [10] and patients [28]. Although not fully understood, the primary effect of cyclodextrin is proposed to mobilize cholesterol out of the late endosomes and lysosomes [10]. Cholesterol transport from late endosomes was shown to regulate the trafficking of soluble N-ethylmaleimide-sensitive factor activating protein receptor (SNARE) protein to the plasma membrane, which affects cargo secretion via the constitutive exocytic pathway [29]. The MβCD-induced mobilization of cholesterol from endosomes/lysosomes may somehow facilitate the secretory mechanism of adiponectin.

Cholesterol egress from late endosomes and lysosomes is mediated by the corporative function of NPC2 and NPC1 [10]. Pharmacological inhibition of NPC1 in adipocytes leads to reduced insulin signaling and insulin-stimulated glucose uptake [23]. By contrast, NPC2 deficiency in adipocytes impairs autophagy and lysosomal activity [30]. To date, whether NPC proteins are involved in adipokine secretion has not been reported. We have observed that the depletion of NPC1 or NPC2 in adipocytes reduced MβCD-induced adiponectin secretion. NPC1 or NPC2 deficiency is known to cause cholesterol accumulation in endosomes/lysosomes, thereby blocking cholesterol egress to other cellular locations, including the plasma membrane and ER. The fact that NPC1 or NPC2 depletion reduced MβCD-induced adiponectin secretion suggests a role of endosomal/lysosomal cholesterol modulation in the vesicle or secretory mechanism of adiponectin.

As an important adipokine that regulates metabolism and insulin sensitivity, understanding the modulating mechanism of adiponectin expression and secretion in adipocytes is important. Adiponectin transcription is positively regulated by insulin and insulin-sensitizing drugs such as thiazolidinediones. Conversely, proinflammatory cytokines attenuate insulin signaling and action, thereby reducing adiponectin transcription [31]. However, the secretory mechanism of adiponectin remains largely unknown. Adiponectin is localized in the Golgi apparatus, and a functional Golgi is required for adiponectin secretion [32,33]. Adiponectin is co-localized with endosome proteins rab 5 and rab 11 and is regulated by rab11 downstream effector proteins [34]. Increasing the pH of lysosomes decreased adiponectin secretion [35], consistent with our results that functional endosomes/lysosomes are involved in MβCD-induced adiponectin secretion. Using sucrose gradient fractionation, we observed that MβCD treatment redistributed adiponectin to lower-density fractions. Although the identity of lower-density fractions requires further characterization, these lower-density fractions may contain endosomes or lysosomes [36]. MβCD treatment may also alter the secretory route of adiponectin to organelles or vesicles closer to the plasma membrane for secretion. In addition to trafficking via the classical endoplasmic reticulum (ER)–Golgi pathway, exosomes have been reported to be involved in adiponectin secretion [37,38]. Interestingly, intracellular cholesterol transport also affects autophagosome–lysosome fusion [22], whereas autophagy function has been reported to regulate adiponectin secretion via exosomes [39]. We have found that cellular cholesterol trafficking via endosomes/lysosomes plays a role in MβCD-induced adiponectin secretion. Whether MβCD treatment also modulates autophagy and exosome formation and their involvement in adiponectin secretion remains to be determined.

BafA1 treatment increased intracellular adiponectin, whereas MβCD treatment did not affect intracellular adiponectin protein. At present, we do not know the reason behind this discrepancy. It is possible that MβCD treatment induced adiponectin secretion, whereas BafA1 inhibition on acidic organelles might also affect protein stability, biosynthesis, or other mechanisms in addition to adiponectin secretion.

The clinical significance of our findings remains to be determined. Although clinical observation favors the idea that cholesterol reduction is beneficial to improving patients with hyperlipidemia, obesity, and diabetes, most of the conclusion comes from the combined effects from different organs, including the liver, adipose tissue, muscle, pancreas, and heart. Moreover, the beneficial effects of cholesterol reduction come from relieving disease conditions such as lipid-loaded or elevated inflammation, which may not be the same as the cholesterol depletion from the untreated control in our cell line model. Therefore, whether induced cholesterol efflux in adipocytes affects adiponectin secretion, thereby improving the metabolic function of patients, merits further investigation. Moreover, specifically inducing cholesterol efflux from endosomes/lysosomes is currently not feasible. Usually, the treatment of MβCD or HDL is applied to trigger intracellular cholesterol efflux. MβCD treatment not only affects cholesterol efflux from endosomes/lysosomes, but also depletes cholesterol in the plasma membrane. Cholesterol depletion in the plasma membrane may also affect membrane integrity and signaling transduction from the plasma membrane. Changes in microdomain integrity or signaling transduction from the plasma membrane may also affect adiponectin secretion. Therefore, developing specific modulators (for example, activators to NPC1/2) may help.

Collectively, our results link the mobilization of cellular cholesterol, especially via the endo/lysosomes, to the mechanism involved in adiponectin secretion in adipocytes. Given that adipocytes constitute the largest cholesterol pool in the human body, handling cholesterol in adipocytes may impact other adipocyte functions, which deserves further investigation.

## 4. Materials and Methods

### 4.1. Materials

Methyl-β-cyclodextrin (MβCD, #C4555) and water-soluble cholesterol (WsCL, #C4951) were purchased from Sigma Chemical (St. Louis, MO, USA). U18666A (#662015) was purchased from Calbiochem (San Diego, CA, USA). Bafilomycin A1 (#11038) was purchased from Cayman Chemical (Ann Arbor, MI, USA).

### 4.2. Cell Culture, Differentiation, and Treatments

3T3-L1 fibroblasts (#CL-173, ATCC, Manassas, VA, USA) were cultured and differentiated into adipocytes as previously described [40]. In brief, cells were grown in the T175 cm^2^ flask to 100% confluency before the induction of differentiation in the medium containing 500 μM isobutylmethylxanthine, 0.2 μM dexamethasone, and 2.5 μg/mL insulin. On day 7 post differentiation, differentiating cells were detached from flasks and re-seeded into dishes or plates for experiments. Experiments were usually carried out between days 12 and 16 post differentiation.

For MβCD treatment, MβCD powder was dissolved directly into the media to the final concentration (4 mM). WsCL and U18666A were prepared in distilled water as a stock of 50 mg/mL and 10 mg/mL, respectively. Bafilomycin A1 was prepared in DMSO as a stock of 1 mM. With the 1 μM final concentration of Bafilomycin A1 in the treatment, 0.1% DMSO was present in the media.

### 4.3. Electroporation of Differentiated Adipocytes

Differentiated 3T3-L1 adipocytes were electroporated with siRNA on day 12 post differentiation as described previously [40]. The siRNA duplexes were designed and custom synthesized from Horizon (Waterbeach, UK): siNPC1, 5′-GAA CAG UAC CUG ACC AUU AdTdT-3′; siNPC2, 5′-UGA AUA AGC UUC CGG UGA AdTdT-3′. Non-targeting siRNA control against Luciferase was described previously [40].

### 4.4. Cholesterol Measurement

The total cell lysate was prepared using RIPA lysis buffer. Cellular cholesterol was determined using the Amplex Red Cholesterol Assay Kit (#A12216, ThermoFisher, Waltham, MA, USA) according to the manufacturer’s instructions. In brief, a 50 μL lysate sample was mixed and incubated at 37 °C for 30 min with 50 μL Amplex Red Reagent. Fluorescence was measured with the excitation and emission at 530 and 590 nm, respectively. A standard was used to calculate cholesterol concentration in the sample. Protein concentrations of cell lysate were measured to normalize cholesterol levels.

### 4.5. Enzyme-Linked Immunosorbent Assay (ELISA)

Adiponectin secretion was measured using the mouse adiponectin ELISA kit (#DY119, R&D systems, Minneapolis, MN, USA) according to the manufacturer’s instructions. Protein concentrations of the cell lysate were measured to normalize adiponectin secretion.

### 4.6. Western Blot Analysis

Western blot analysis was performed as described previously [40]. Twenty micrograms of total protein was loaded per lane in the SDS-PAGE gel. Anti-adiponectin antibodies (#2789) were purchased from Cell Signaling Technology (Beverly, MA, USA). Anti-α-tubulin (#T5168) and NPC2 (#HPA000835) antibodies were from Sigma Chemical (St. Louis, MO, USA). Anti-NPC1 (#ab134113) antibodies were from Abcam (Cambridge, UK).

### 4.7. Lipid Raft Fractionation

Lipid raft fractionation was performed as described previously [16]. Cells from two ten-centimeter dishes were collected and centrifuged at 100× *g* for 5 min. The cell pellet was resuspended in 1 mL TNE/Triton X-100 buffer (25 mM Tris-HCl, pH 7.4, 150 mM NaCl, 5 mM EDTA, and 1% Triton X-100) and was incubated on ice for 20 min. The lysate was homogenized and then brought to 40% sucrose with an equal volume of 80% sucrose in an ultracentrifuge tube. The lysate was layered with a linear sucrose gradient (5 to 30%) and centrifuged in a Beckman SW41 rotor at 200,000× *g* at 4 °C for 16 h. Each 1 mL fraction from the top to the bottom was collected.

### 4.8. RNA Analysis

RNA preparation and quantitative polymerase chain reaction (qPCR) were performed as described previously [40]. Gene expression was calculated using the ΔΔCT method after normalization to the housekeeping gene 36B4 (*Rplp0*). The following primers were used for qPCR: *Adipoq* forward 5′-TGT TCC TCT TAA TCC TGC CCA-3′, reverse 5′-CCA ACCTG CAC AAG TTC CCT T-3′; *Rplp0* forward 5′-GCG ACC TGG AAG TCC AAC TAC-3′, reverse 5′-ATC TGC TGC ATC TGC TTG G-3′.

### 4.9. Statistics

Data were analyzed was performed using InStat 3.0 (GraphPad Software Inc., San Diego, CA, USA). Statistical significance was evaluated using unpaired Student’s two-tailed *t*-test or one-way ANOVA with the Student–Newman–Keuls post hoc test for multiple comparisons (* *p* < 0.05, ** *p* < 0.01, *** *p* < 0.001). All data are presented as the mean of at least three experimental replicates with the standard error of the mean (SEM).

## Figures and Tables

**Figure 1 ijms-24-14718-f001:**
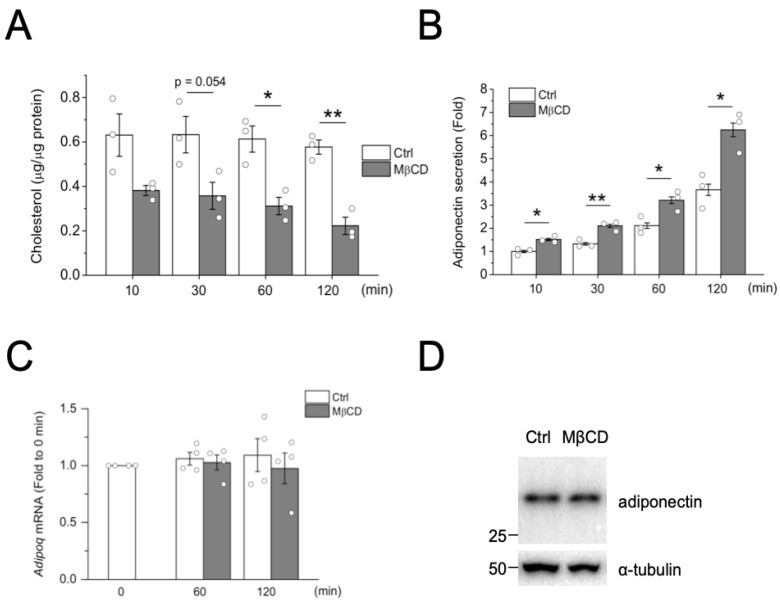
Depletion of cholesterol induces adiponectin secretion in adipocytes. Differentiated 3T3-L1 adipocytes were either untreated (Ctrl) or treated with 4 mM MβCD in duplicates for 10, 30, 60, or 120 min. (**A**) Cellular cholesterol levels were measured as described in Section 4. (**B**) Adiponectin secretion into the media was measured using ELISA. Data are expressed as means ± S.E. from three independent experiments. * *p* < 0.05, ** *p* < 0.01. (**C**) Differentiated 3T3-L1 adipocytes were either untreated (Ctrl) or treated with 4 mM MβCD for 60 or 120 min. Adiponectin mRNA (*Adipoq*) was measured using qPCR. Relative expression was compared to 0 min control. Data are expressed as means ± S.E. from four independent experiments. (**D**) Differentiated 3T3-L1 adipocytes were either untreated (Ctrl) or treated with 4 mM MβCD for 120 min. The cell lysate was subjected to Western blot analysis using anti-adiponectin antibodies.

**Figure 2 ijms-24-14718-f002:**
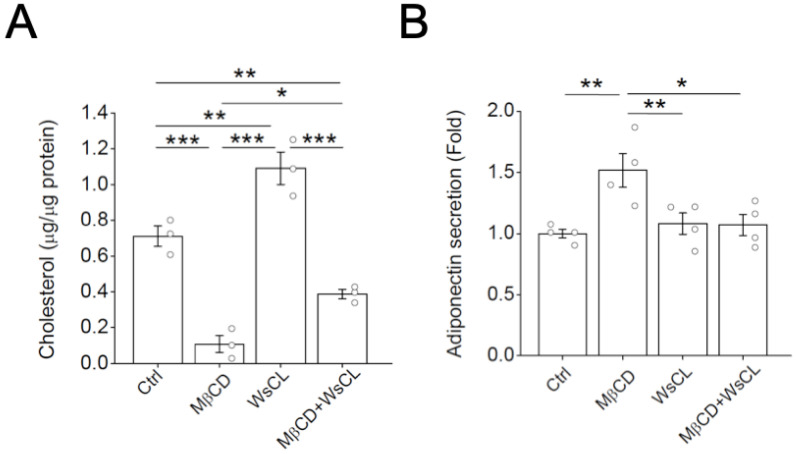
Depletion, but not the addition of cholesterol, modulated adiponectin secretion in adipocytes. Differentiated 3T3-L1 adipocytes were untreated (Ctrl) or treated with 4 mM MβCD, 250 μg/mL water-soluble cholesterol (WsCL), or both (MβCD + WsCL) in duplicates for 2 h. (**A**) Cellular cholesterol levels were measured as described in Section 4. (**B**) Adiponectin secretion into the media was measured using ELISA. Data are expressed as means ± S.E. from at least three independent experiments. * *p* < 0.05, ** *p* < 0.01, *** *p* < 0.001.

**Figure 3 ijms-24-14718-f003:**
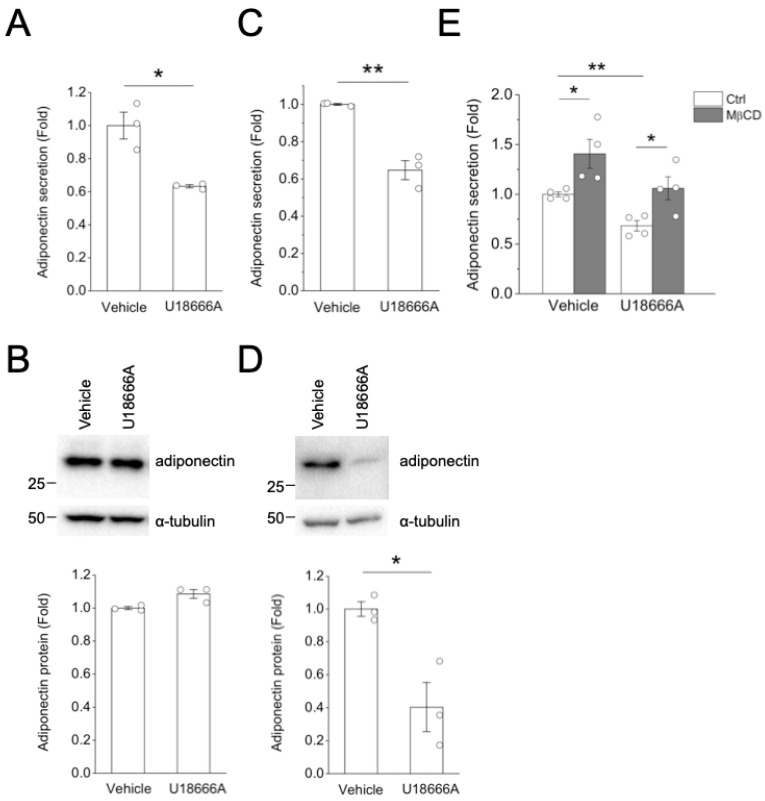
U18666A treatment reduced basal and MβCD-induced adiponectin secretion. (**A**,**B**) Differentiated 3T3-L1 adipocytes were treated in duplicates with Vehicle (distilled water) or 10 μg/mL U18666A for 2 h. (**A**) Adiponectin secretion into the media was measured using ELISA. * *p* < 0.05. (**B**) The cell lysate was subjected to Western blot analysis using anti-adiponectin antibodies. Data from three independent experiments are shown. (**C**,**D**) Differentiated 3T3-L1 adipocytes were treated in duplicates with Vehicle (distilled water) or 10 μg/mL U18666A for 24 h. (**C**) Adiponectin secretion into the media was measured using ELISA. ** *p* < 0.01. (**D**) The cell lysate was subjected to Western blot analysis using anti-adiponectin antibodies. Data from three independent experiments are shown. * *p* < 0.05. (**E**) Differentiated 3T3-L1 adipocytes were untreated (Ctrl) or treated with 4 mM MβCD, with Vehicle (distilled water) or 10 μg/mL U18666A for 2 h. Experiments were performed in duplicates. Adiponectin secretion into the media was measured using ELISA. Data from four independent experiments are shown. * *p* < 0.05, ** *p* < 0.01.

**Figure 4 ijms-24-14718-f004:**
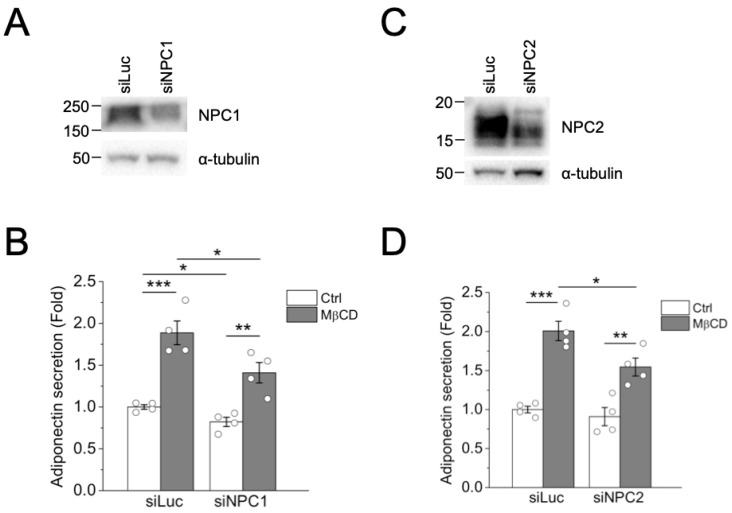
Depletion of NPC1 or NPC2 reduced MβCD-induced adiponectin secretion in adipocytes. (**A**,**B**) Differentiated 3T3-L1 adipocytes were electroporated with non-targeting luciferase siRNA (siLuc) or siRNA against NPC1 (siNPC1). At 24 h post transfection, cells were untreated (Ctrl) or treated with 4 mM MβCD for 2 h. (A) NPC1 depletion was verified through Western blot analysis. (**B**) Adiponectin secretion was determined using ELISA. Experiments were performed in duplicates. Data are expressed as means ± S.E. from four independent experiments. * *p* < 0.05, ** *p* < 0.01, *** *p* < 0.001. (**C**,**D**) Differentiated 3T3-L1 adipocytes were electroporated with non-targeting luciferase siRNA (siLuc) or siRNA against NPC2 (siNPC2). At 24 h post transfection, cells were untreated (Ctrl) or treated with 4 mM MβCD for 2 h. (**C**) NPC2 depletion was verified through Western blot analysis. (**D**) Adiponectin secretion was determined using ELISA. Experiments were performed in duplicates. Data are expressed as means ± S.E. from four independent experiments. * *p* < 0.05, ** *p* < 0.01, *** *p* < 0.001.

**Figure 5 ijms-24-14718-f005:**
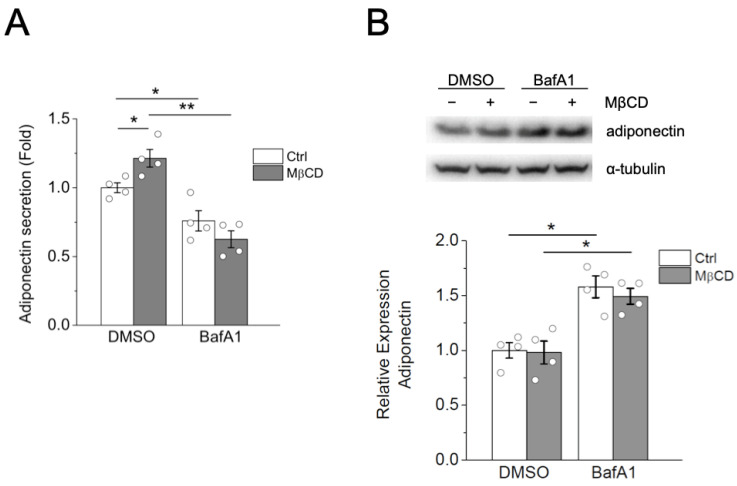
Bafilomycin A1 treatment attenuated MβCD-induced adiponectin secretion. (**A**,**B**) Differentiated 3T3-L1 adipocytes were untreated (Ctrl) or treated with 4 mM MβCD, together with DMSO (DM) or 1 μM bafilomycin A1 (BafA1), for 2 h. (**A**) Adiponectin secretion into the media was measured using ELISA. Experiments were performed in duplicates. Data are expressed as means ± S.E. from four independent experiments. * *p* < 0.05, ** *p* < 0.01. (**B**) The cell lysate was subjected to Western blot analysis using anti-adiponectin and BECN1 antibodies. Data are expressed as means ± S.E. from four independent experiments. * *p* < 0.05.

**Figure 6 ijms-24-14718-f006:**
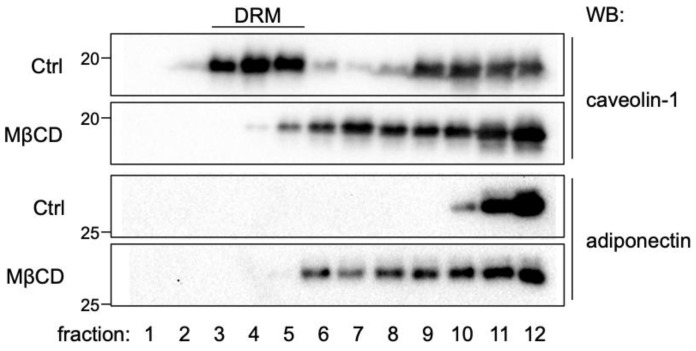
MβCD treatment alters the distribution of adiponectin in lipid raft fractionation. 3T3-L1 adipocytes were either untreated (Ctrl) or treated with 4 mM MβCD for 2 h. Cell lysates were subjected to sucrose density-gradient ultracentrifugation. The levels of caveolin-1 and adiponectin in each fraction were detected through Western blot analysis. DRM, detergent-resistant membrane.

**Figure 7 ijms-24-14718-f007:**
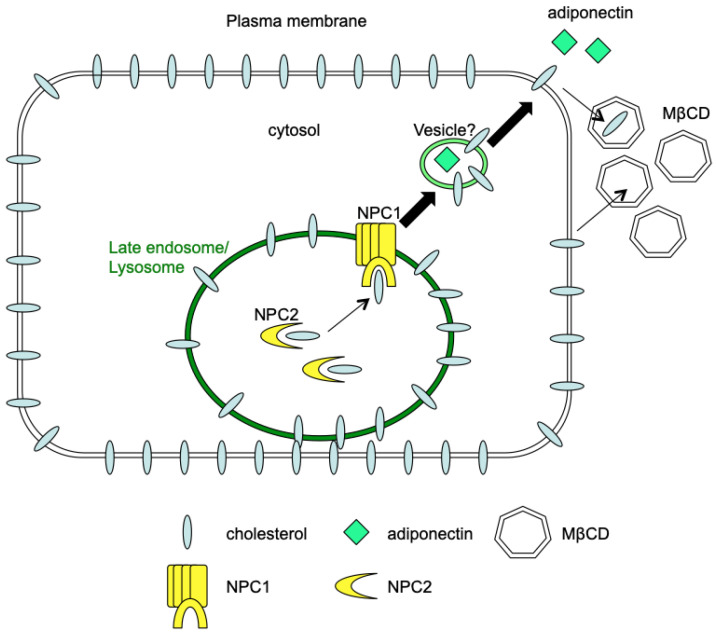
Schematic diagram of MβCD-mediated cholesterol depletion and adiponectin secretion in adipocytes. MβCD, with a high affinity for cholesterol, depletes cholesterol from the plasma membrane, which subsequently induces cholesterol efflux and depletion of intracellular cholesterol such as those in the late endosomes and lysosomes. The increased efflux of cholesterol from late endosomes/lysosomes, mediated by NPC proteins, may facilitate the yet-to-be-characterized secretory mechanism (for example, vesicles) of adiponectin, leading to increased secretion of adiponectin from adipocytes.

## Data Availability

Data available on request. The data presented in this study are available on request from the corresponding author.

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
