# Peer review of "Cyclodextrin-Mediated Cholesterol Depletion Induces Adiponectin Secretion in 3T3-L1 Adipocytes"

_ijms, 2023, doi:10.3390/ijms241914718_

Round 1

Reviewer 1 Report

After reviewing your manuscript I must say that I find it highly informative and well executed. Your research sheds light on how cholesterol in adipocytes can be influenced and its impact on adiponectin secretion. The insights you provide into the mechanisms involved are truly valuable.

The study design was well thought out. The methods used for manipulating cholesterol analyzing adiponectin secretion and fractionating lipid rafts seemed reliable. I found your discoveries regarding the increase in adiponectin secretion following MβCD treatment well as the role played by NPC proteins in this process to be particularly intriguing. Moreover your use of siRNA to deplete NPC1 and NPC2 was a step in establishing their involvement, in cholesterol trafficking.

I have a suggestions to enhance the clarity and completeness of the manuscript:

1. Can you provide information about the experimental setup for the MβCD treatment? It would be helpful to know the duration of treatment and the specific concentration of MβCD used.

2. Could you elaborate on why WsCL was combined with MβCD to restore cholesterol levels? It would be beneficial to understand the rationale behind restoring cholesterol levels. How it relates to the observed effects on adiponectin secretion?

3. In the Discussion section it would be valuable if you could expand on the implications of redistributing adiponectin to lower density fractions after MβCD treatment. Discussing its significance in relation to adiponectin secretion and cellular cholesterol modulation would provide an understanding.

These revisions will contribute towards improving both clarity and comprehensibility of your manuscript.

Reviewer 2 Report

The study from Chiang et al. examined the effects of methyl-β-cyclodextrin (MβCD) on cholesterol depletion and adiponectin secretion in the 3T3-L1 adipocytes.

Here my comments related with this manuscript.

-The title should be more specific, considering methyl-β-cyclodextrin and the word modulation should be changed to depletion.

-In the abstract section, please add the concentrations of methyl-β-cyclodextrin (MβCD), U18666A, and bafilomycin A1 used for the treatments of differentiated 3T3-L1 adipocytes. With respect to the following phrase: “Finally, MβCD treatment altered cellular distribution of adiponectin.” It should be clearer. In addition, the next phrase should be restructured “Our results show that modulation of cellular cholesterol regulates the secretion of adiponectin,” changing the word modulation by depletion and the word regulates by elevates..

-In the introduction section, in the page 1 and line 33: “Adipocytes are known to store a significant amount of free, ….” The phrase seems incomplete.

-Please add a reference for the following phrase (page 2, line 50-52): “Defective cholesterol trafficking………….. highlighting the importance of cholesterol modulation in cell function.” Also a reference for the phrase (page 2, line 59-60): “Moreover, adiponectin reduces the uptake of oxidized low-density lipoprotein and inhibits foam cell formation, thereby suppressing the development of atherosclerosis.”

-In the materials and methods section, the authors should add more details about culture of 3T3-L1 adipocytes, considering amount of cells, cell culture dishes, details for treatments of 3T3-L1 adipocytes for all different experiments, including DMSO percentage used. With respect to western blot analysis, please add the amount of total protein per lane in SDS-PAGE gel. In addition, please add if experiments were done in triplicate.

-In the results section, the authors should describe the results in this section with the statistically significant values. Why the expression of adiponectin levels were not expressed in units? In the figure 3, please homologate DMSO with DM and U18666A with U, and Figures B and D need a western blot histogram, respectively. Why 10 μg/ml U18666A were used in this study, please add a reference. Please in all western blots add molecular weight of the proteins analyzed. In the figure 5B, histograms are different with respect to band intensities, please check them. With respect to western blots as supplements, what do the bands that do not have a legend indicate? And some gels do not have molecular weight marker.

-I suggest to analyzing the BECN1 protein related to autophagy by western blot (Figure 5B).

-A schematic diagram is needed to illustrate the effects of methyl-β-cyclodextrin on adiponectin expression.

-The acronym SNARE should be defined, and some phrase are very long, please use commas or semicolons.

-Please review the following phrases because they are difficult to understand: page 7 and line 234-236 “Disruption of lipid rafts in adipocytes may reduce insulin signaling [22, 23] but induce proinflammatory signaling [12], both may affect leptin expression in adipocytes.

However, there is another phrase (page 8, line 265-267): “Adiponectin transcription is positively regulated by insulin and insulin-sensitizing drugs such as thiazolidinediones, but is negatively regulated by proinflammatory cytokines [26].”

First phrase: the disruption of lipid rafts in adipocytes induce proinflammatory signaling. However, adiponectin transcription is negatively regulated by proinflammatory cytokines. Please explain the possible effects of methyl-β-cyclodextrin on adiponectin expression.

-The authors should add study limitations in the manuscript.

There are some grammatical errors and some typos. 

Round 2

Reviewer 2 Report

I have no comments. 

The English Language is fine, minor editing of English language required.

Author Response

Thank you for the comment. The manuscript has been checked by a colleague fluent in English writing and is revised accordingly.

Reviewer 3 Report

The authors have satisfactorily addressed the comments. Please refer to my additional comments below:

1. Referring to the authors' response to comment no. 3, could they put the explanation from the 1st paragraph (As described above, MβCD, with its high affinity to cholesterol, depletes cholesterol from plasma membrane and also intracellular compartments when cells are incubated with MβCD. Therefore, MβCD treatment affected both plasma membrane cholesterol and intracellular cholesterol. We are sorry for not describing clearly and causing the confusion.) in the revised manuscript? Or refer to the intracellular cholesterol as simply cellular cholesterol?

2. Could the authors please put the response to comment no. 9 in the discussion? (BafA1 treatment increased intracellular adiponectin, whereas MβCD treatment did not affect intracellular adiponectin protein. At presence we do not know the reason behind this discrepancy. It is possible that MβCD treatment induced adiponectin secretion, whereas BafA1 inhibition on acidic organelles might also affect protein stability, biosynthesis, or other mechanism in addition to adiponectin secretion.)

3. Could the authors please explain the clinical significance of this finding? What is the importance of cholesterol depletion from adipocytes in the metabolic conditions such as hyperlipidemia, obesity and diabetes? How can the cholesterol efflux be triggered in patients to improve their adiponectin secretion, which can be beneficial to improve insulin sensitivity? If the cholesterol depletion causes elevation of pro-inflammatory signaling, how is it beneficial/harmful to modulate cholesterol trafficking in adipocytes? Any substantial explanation of the clinical significance of the findings of this study could be helpful.

Round 3

Reviewer 3 Report

The authors have made necessary changes to the manuscript. Please include the response to my comment no. 3 in the second round of review in the discussion, by making appropriate changes to make it more concise. 
